Resource

# A broken network of susceptibility genes in the monocytes of Crohn's disease patients

Hankui Liu[1,2], Liping Guan[1,2,4], Xi Su[2,4], Lijian Zhao[1,2,6], Qing Shu[3], Jianguo Zhang[1,5,6]

**Genome-wide association studies have identified over 200 genetic loci associated with inflammatory bowel disease; however, the mechanism of such a large amount of susceptibility genes remains uncertain. In this study, we integrated bioinformatics analysis and two independent single-cell transcriptome datasets to investigate the expression network of 232 susceptibility genes in Crohn's disease (CD) patients and healthy controls. The study revealed that most of the susceptibility genes are specifically and strictly expressed in the monocytes of the human intestinal tract. The susceptibility genes established a network within the monocytes of health control. The robustness of a gene network may prevent disease onset that is influenced by the genetic and environmental alteration in the expression of susceptibility genes. In contrast, we showed a sparse network in pediatric/adult CD patients, suggesting the broken network contributed to the CD etiology. The network status of susceptibility genes at the single-cell level of monocytes provided novel insight into the etiology.**

## Introduction

Crohn's disease (CD) and ulcerative colitis (UC), which are usually referred to as inflammatory bowel disease (IBD), are a group of complex disorders characterized by chronic relapsing intestinal inflammation (1). The twin cohort estimated an average heritability of 0.75 for CD and 0.67 for UC, indicating genetic factor plays a critical role in IBD (2). Thus far, genome-wide association studies (GWAS) have identified more than 200 loci associated with IBD (3, 4, 5, 6). Most of the susceptibility loci of IBD share a consistent effect in CD and UC and in population of both European and non-European descent (6). The common single nucleotide polymorphisms (SNPs) in a GWAS of up to 86,640 European descent and 9,846 individuals of non-European descent contributed a heritability of 21% for CD and 27% for UC (7). A significant set of 163 loci in a GWAS of 75,000 cases and controls explained a disease variance (variance being subject to fewer assumptions than heritability (8)) from 8.2% to 13.6% in CD

and from 4.1% to 7.5% in UC (9). Overall, the presence of missing heritability indicated a significant lack of understanding of the causes of IBD. Because the GWAS strategy hypothesized that loci are independent (10), the absent heritability could be due to gene–gene interactions (8, 11), as well as the causes of incomplete linkage disequilibrium between causal variants and genotyped SNPs (12), missing variant with small effect (11), missing rare variant with large effect (11), structure variant poorly captured by existing arrays (11), and epigenetic modifications (13). Mesbah-Uddin et al constructed a gene interaction network using co-expression and protein–protein interaction data, and discovered 11 clusters (14). Jimmy et al implicated 38 novel loci and linked 36 genes located within the novel loci to the established network of susceptibility genes (6). Lauren et al constructed a network model using functional annotations and predicted 12 key driver genes that modulated the network regulatory states (15). These studies demonstrated connections between susceptibility genes at the level of bulk cells or individuals, and the gene interactions may have a role in IBD.

Benefited by single-cell RNA (scRNA)-sequencing technology (16), we can investigate the gene expression characteristics, disease-related cell types, and gene networks at the single-cell level. Our previous research (17, 18) in this field indicated that there are specific cell types associated with genes and diseases. For instance, serotonin neurons in the brain express genes associated with anxiety disorders, whereas motor neurons in the spinal cord express genes associated with amyotrophic lateral sclerosis. IBD was primarily mediated by $T_h1$ cells in CD[20], $T_h2$ cells in UC (19), and $T_h17$ cells in both CD and UC (20). A recent scRNA study in the human intestinal tract indicated additional cell types (dendritic cells, monocytes, innate lymphoid cells) that enriched the expression of IBD susceptibility genes (21). These results suggested a significant connection between susceptibility genes, specific cell type, and related disease. To address the network of susceptibility genes, we firstly identified the IBD-related cell type from the cellular landscape of the human intestinal tract. Subsequently, we constructed a graph of gene expression in the specific cell type and displayed vastly different network statuses for pediatric/adult CD patients and healthy controls.

---

[1]Hebei Industrial Technology Research Institute of Genomics in Maternal & Child Health, Clin Lab, BGI Genomics, Shijiazhuang, China    [2]BGI Genomics, Shenzhen, China    [3]Department of Gastroenterology, The First Affiliated Hospital of Shenzhen University, Shenzhen Second People's Hospital, Shenzhen, China    [4]Department of Biology, University of Copenhagen, Copenhagen, Denmark    [5]BGI Research, Shenzhen, China    [6]Hebei Medical University, Shijiazhuang, China

Correspondence: zhangjg@genomics.cn; sq6060@163.com; zhaolijian@genomics.cn

# Results

## Monocytes enriched the expression of IBD susceptibility genes

We extracted 232 IBD susceptibility loci from the most recent and comprehensive GWAS (6) and then annotated 232 susceptibility genes. To address the cellular basis of these genes, we calculated the cell-type specificity of the expression of susceptibility genes in the single-cell transcriptome data of 7 pediatric and 46 adult CD patients (Table 1). The results exhibited that monocytes significantly enriched the expression of IBD susceptibility genes in pediatric/adult CD (Fig 1A), indicating an involvement of monocytes in CD pathogenesis. The number of monocytes was obviously (Fisher's exact test, $P$ = 2.7 × $10^{-108}$) increased in CD patients (Fig 1B), which is consistent with previous research (22). Monocytes are a part of the body's first line of defense. These cells eliminate pathogens through phagocytosis or by emitting a wide variety of inflammatory mediators that influence the intestinal immune system (23). Besides monocytes, we showed an involvement of T helper cells in CD (i.e., $T_h1$ in pediatric CD, $T_h17$ in adult CD). T helper cells, known as CD4-positive cells, are a main driver of IBD (23). The accumulation of $T_h1$ cells in the intestinal tract of the CD patient is directly linked to the disease (24). $T_h17$ cells, identified by *IL17A*-positive cells, harbored significant enrichment (Fig 1A) of IBD gene expressions in adult CD and a suggestive significance (raw $P$ = 0.008; false discovery rate [FDR] = 0.12) in pediatric CD. $T_h1$ cells were not identified in the adult sample of Kong et al's study (25). The association of ILC3 (type 3 innate lymphoid) cells in pediatric did not present in adult CD, which could be attributed to the fact that ILC cells play a role in the initial phase of the disease (26). ILC3 cells are the innate counterparts of $T_h17$ cells (27). These cells are responsible for mucosal homeostasis in the gastrointestinal tract and contribute to the progression and exacerbation of IBD (28).

## IBD susceptibility genes are dosage-sensitive in monocytes

We have determined that most of the susceptibility genes are specifically expressed in monocytes; however, it remains uncertain how these genes influence CD. Jimmy et al showed 12 of 38 IBD-related SNPs are expression quantitative trait loci (eQTL) (6). Yukihide et al also showed that 63 of 200 IBD risk loci are eQTL (29).

These results suggest the modified expression of susceptibility genes may influence the disease onset. In order to determine the tolerance of these genes to the modification, we evaluated the dosage requirement for a specific cell type using the strictness measure. Strictness refers to the spectrum of expression fold change in normal cells. Our prior research demonstrated that amyotrophic lateral sclerosis susceptibility genes with a known mechanism of loss of function (LoF) exhibit high strictness and the gain-of-function genes exhibit low strictness (18). On this premise, we calculated the strictness of susceptibility genes in monocytes and demonstrated that susceptibility genes have significantly higher strictness than LoF-tolerant (LoFT) genes ($P$ = 5.52 × $10^{-15}$) (Fig 1C) and all background genes ($P$ = 1.13 × $10^{-5}$) (Fig 1D), indicating that they are dosage-sensitive. In addition, we also employed the Z-score of LoF as specified by the gnomAD publication (30). The distributions exhibited that the LoF Z-scores of IBD susceptibility genes are greater than those of LoFT genes (Fig 1E). This result is consistent with that of the strictness measure. In light of these findings, we concluded that most of the susceptibility genes influence disease/phenotypes through dosage regulation.

## Connections between susceptibility genes are broken in monocytes of CD patients

Now, we have indicated that the susceptibility genes are specially expressed in monocytes and their gene activities are carried out through the control of their dosage. On the basis of these two findings, we constructed a gene expression network in monocytes using a Gaussian graphical model (GGM) to investigate the relationships between susceptibility genes. Briefly, we observed a different density of the graph between pediatric CD patients and healthy controls (Fig 2A). This difference was validated in the additional single-cell transcriptome dataset of 46 adult CD patients and 25 healthy controls (Fig 2B).

The network graph revealed 138 connections between 81 genes in healthy adults, 100 connections between 74 genes in healthy pediatric controls, 73 connections between 86 genes in fetal controls, and 30 connections between 57 genes in pediatric CD (Fig 3A). Of the 283 unique connections, 79 presented protein–protein interactions from the STRING database (31). This significant proportion of overlap indicated these connections are reliable

**Table 1. Summary of public single-cell transcriptome data.**

| Study | Major tissue | Cell number | Cell type | Case | Control | Resource |
|---|---|---|---|---|---|---|
| Elmentaite et al | Terminal ileum and colon | 428,000 | 6 main cell types | Pediatric CD [N = 7, age = 9–14] | Pediatric [N = 8, age = 4–10] | https://www.gutcellatlas.org |
| | | | 133 subtypes/states | | Adult [N = 6, age = 20–75] | |
| | | | | | Fetal [N = 16, age = 6.7–17 Wk] | |
| Kong et al | Terminal ileum and colon | 720,633 | 3 main cell types | Adult CD [N = 46, age = 20–74] | Adult [N = 25, age = 25–74] | https://cellxgene.cziscience.com/datasets |
| | | | 54 subtypes/states | | | |

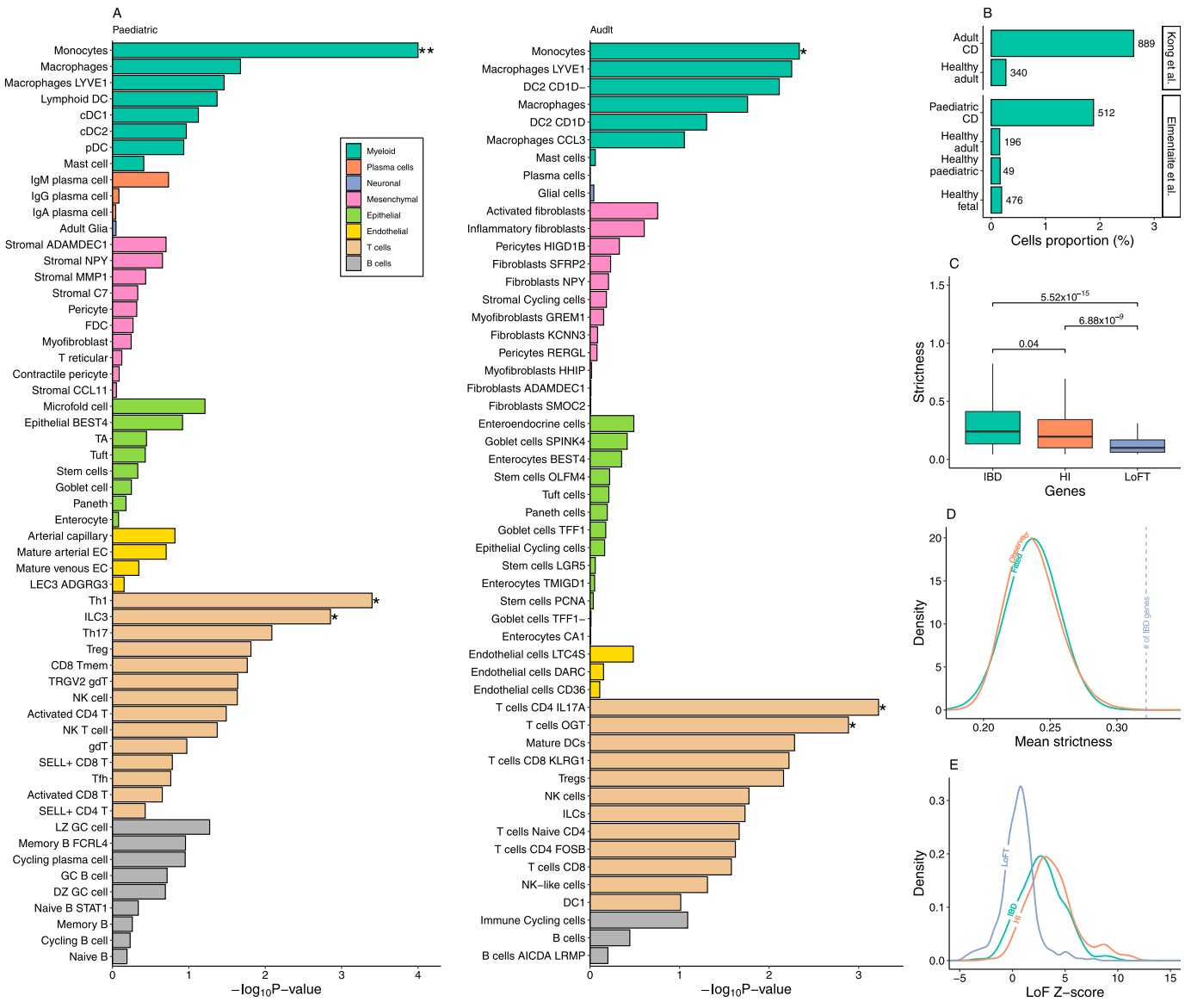

**Figure 1. Expression characteristic of IBD-related genes.**
**(A)** Histograms refer to the raw *P*-values of enrichment in different cell types and states of the human gut. The * refers to significant enrichment after FDR adjustment (FDR < 0.05). Cells are ranked according to their enrichment of susceptibility genes. **(B)** Compared with the healthy controls, the proportion of monocytes in CD patients is significantly increased. The y-axis refers to the proportion of cells. The number on the top of the histogram refers to the number of cells. **(C)** Strictness of IBD susceptibility gene expressions was compared with the haploinsufficient (HI) and loss-of-function–tolerant (LoFT) genes in monocytes. **(D)** Mean strictness of IBD susceptibility genes is displayed at a blue dash line against the observed distribution (red) and fitted distribution (green) of background genes. **(E)** Distributions of LoF Z-scores were differed by IBD susceptibility genes, HI genes, and LoFT genes.

(binomial test, $P = 3.23 \times 10^{-30}$). The density and cluster of a network of susceptibility genes are lower in pediatric CD patients compared with healthy pediatric controls (graph density: 0.019 versus 0.037; clustering coefficient: 0.098 versus 0.360) (Fig 3A). The differences in the network status (edge, density, clustering coefficient) were confirmed in the comparison of adult CD patients and healthy adults (Fig S1A). The gene degrees of network in pediatric CD are significantly less than those in healthy pediatric controls ($P = 2.4 \times 10^{-6}$) (Fig 3B), as well as those between adult CD (inflamed colon) and healthy adult ($P = 0.0005$) (Fig S1B). These results indicated that the network of susceptibility genes in CD patients is disrupted.

Using a threshold of degree > 5 and central score > 180, we identified 2, 5, and 8 central genes in the fetal, pediatric, and adult healthy networks, respectively (Fig 3C). However, only one central gene was identified in the network of CD patients. The key central gene present in the four networks is *RPL3*, which is marked by rs12627970. The gene adjacent to this locus was annotated as *SYNGR1* in the previous study (32). Using a similar threshold of degree > 4 and central score > 50, we identified the key central gene, *RPL3*, in the network of adult CD patients (Fig S1C).

The aforementioned findings suggested a possible correlation between the network status and disease status. To determine

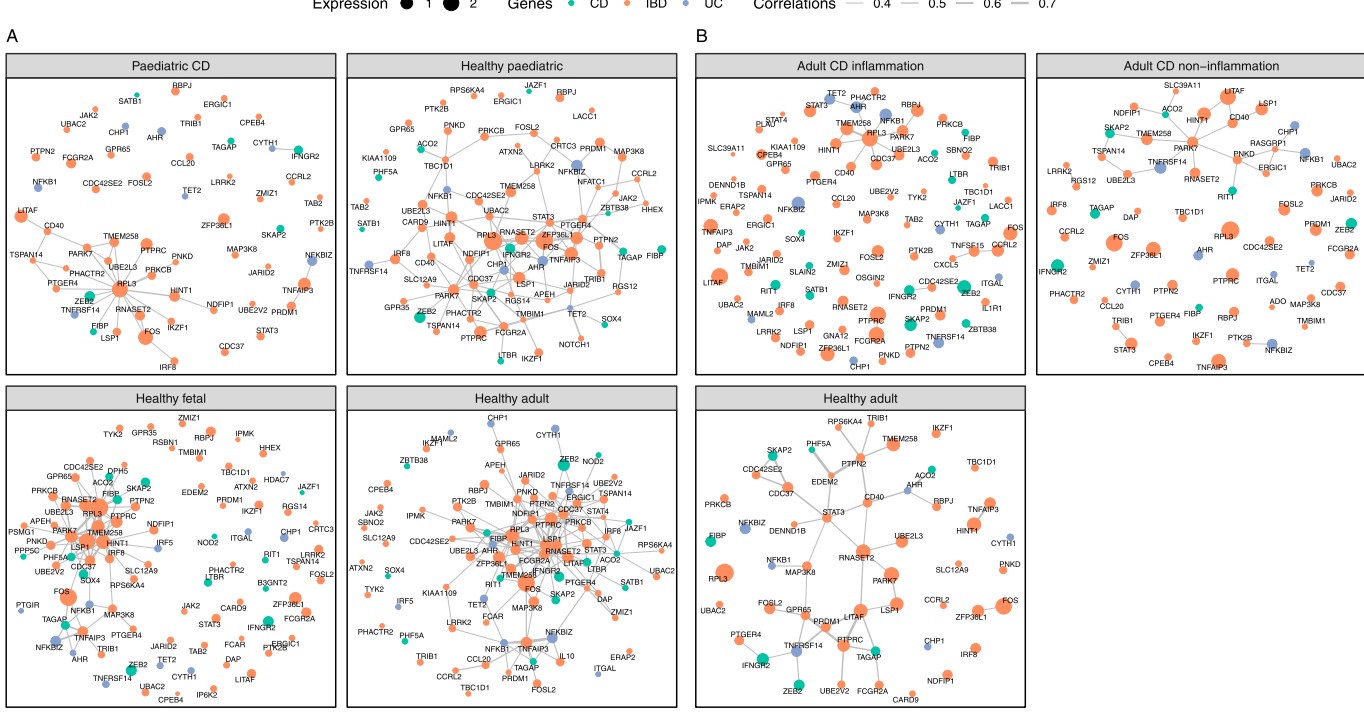

**Figure 2. Gene-gene interaction network.**
**(A, B)** Expression networks of IBD susceptibility genes in monocyte single-cell transcriptomes of pediatric CD (A), adult CD (B), and healthy controls (A, B). Color points refer to the genes related to IBD, CD, and/or UC, separately. The edge size refers to Pearson's correlation coefficient of the expression of two linked genes. The node size refers to the mean gene expression.

whether the aberrant network is a cause or a result of the inflammation, we conducted a comparison of the network status between the inflamed colon and the non-inflamed colon of CD patients (Fig 2B). Statistical analysis (*P* = 0.144) does not support a difference in network density between the inflamed colon and non-inflamed colon (Fig S1B). Considering the notable disparity (*P* = 0.01) in network density between the healthy controls and the non-inflamed colon of CD patients (Fig S1B), we proposed that the disrupted network contributes to the development of CD. We have to mention that the status of inflammation also impacts the network status. A reduction in edge/density/clustering coefficient was seen when comparing the non-inflammation and inflammation network, indicating the network status was severely disrupted in the inflamed colon.

We partitioned the entire network into subgraphs using hierarchical clustering (Fig 4). The phylograms exhibited that a key central gene (*STAT3*) in healthy pediatric controls/adults (left) was disconnected in pediatric/adult CD patients (right), resulting in the formation of individual genes in the broken network. We performed a KEGG pathway enrichment analysis at the six individual genes (*PTPN2*, *IFNGR2*, *ZBTB38*, *LRRK2*, *UBAC2*, and *CHP1*) that were supposed to directly connect with *STAT3*. The result showed six pathways significantly (FDR < 0.01) enriched these genes (Fig S2). Jak/STAT signaling pathway (33), HIF-1 signaling pathway (34), and T_h17 cell differentiation (35) were known to play an important role in IBD. Over the last decade, programmed death-ligand 1 (PD-L1) expression was proposed as a key mechanism for the mucosal

tolerance in the gut (36). Recently, a study suggested that chronic Toxoplasma gondii infection enhances monocyte activation to increase inflammation associated with a secondary environmental insult (37).

# Discussion

GWAS have reported a large number of disease susceptibility genes (38). How such a large number of susceptibility genes affect disease phenotypes remains a fundamental question in the genetics field. The polygenic score method showed genes contributed to disease may follow an additive model that sums the disease risk (39). However, the GWAS common SNPs only contribute 25% of IBD heritability. Moreover, the significant SNPs contribute a part of the heritability explained by all common SNPs (6, 7). Severe mutations contribute a large effect to very early-onset IBD but are rare in population (40). Besides independent risk, gene–microbiota interactions and gene–gene interactions may have a profound effect on IBD pathogenesis. Chu et al described that *ATG16L1*/*NOD2* and microbiome cooperate to promote beneficial immune responses (41). Aleknonytė-Resch et al revealed a significant interaction between rs26528 in the *IL27* gene and rs9297145 in the *KPNA7* gene (42). At the gene level, we showed that the expression network status of susceptibility genes contributes to CD. A key feature of the network is robustness that is tolerant to genetic/environmental attacks or

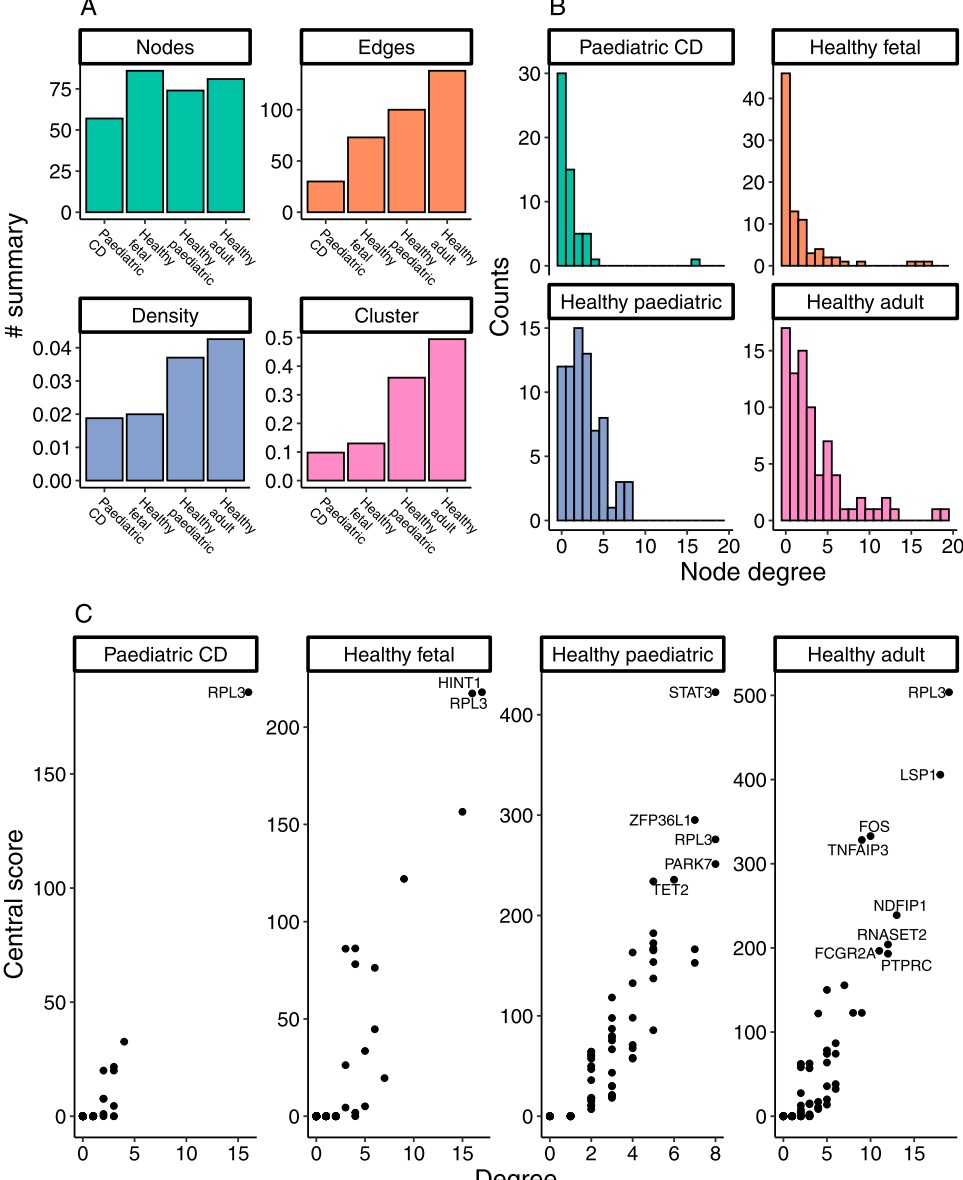

**Figure 3. Network metrics.**
**(A)** Statistical summary of the network graph. The node refers to the number of genes in the network. The edge refers to the number of connections between genes. The density refers to the density of edges among nodes. The cluster refers to the clustering coefficient of the network. **(B)** Degree refers to the number of edges linked to a node. **(C)** Node degree and central score are used for indicating the central gene in a subgraph. Thresholds of degree > 5 and central score > 180 were used for identifying central genes.

random fluctuations in gene expression (43). Marigorta et al showed that SNPs with eQTL effects modified gene expression significantly and the transcriptional risk score outperformed genetic risk scores in estimating IBD risk (32). This result indicates that the disease is influenced by common SNPs through changes in gene expression. The gnomAD database documented LoF variants where an allele results in a reduction of half the dosage. We calculated the cumulative allele frequency (CAF) of LoF variants at 43 susceptibility genes with eQTL (Table S1). The estimated CAF is 0.1, indicating that there is a probability of 0.1 for a person to possess at least one LoF allele at the 43 susceptibility genes. What are the reasons for the prevalence of such a common LoF CAF in a population? Moreover, our research and previous studies (6, 29) demonstrated that most of the IBD susceptibility genes are dosage-sensitive. However, few studies have reported causal mutations at these susceptibility

genes in pedigrees (44). Collectively, the observation of common LoF CAF and rarely reported causal mutation are opposed to the theory of gene dosage contributed to IBD. Notably, the gene expression network in the healthy controls can be an explanation. In contrast, the gene–gene interactions were disconnected in CD patients and the network graph was split into multi-independent subgraphs. Taken together, we propose a possible mechanism that in the broken network of patients, the dosage-sensitive genes altered by eQTL cannot be regulated by the network center and/or compensated by correlated genes.

Genes in the network center are key regulators that may play important roles in IBD. We identified the numbers of central regulators, such as *RPL3*, *STAT3*, *PAPK7*, and *RNASET2* in the healthy network. These genes are known to play important roles in the regulation of IBD-related functions. *STAT3* is a mediator gene that

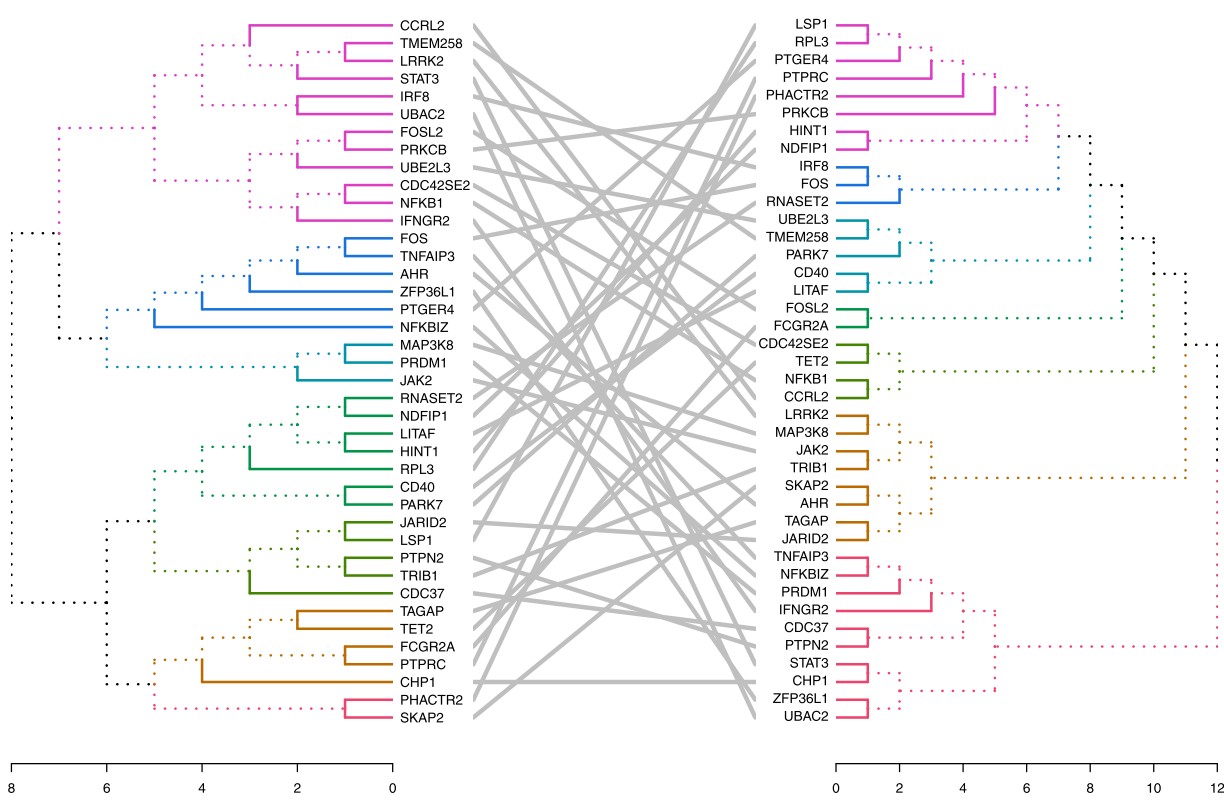

**Figure 4. Network subgraph comparison.**
Subgraphs were classified by the hierarchical clustering algorithm. Polynemes were used to display the subgraphs of the network by edge colors. Gene links were used to compare the subgraph differences between pediatric health controls (left) and pediatric CD (right).

regulates innate and adaptive immunity (45). The loss of *STAT3* in immune cells caused severe inflammation (45). *PARK7* was shown to regulate the IBD-related inflammation in vitro and in vivo (46). *RNASET2* expression decreases in response to T-cell activation (47). The overexpression of *RNASET2* significantly reduced IFN-γ secretion (47). The *STAT3* and *RNASET2* were suggested to be potential therapeutic targets in the treatment of IBD (47, 48), as well as *PARK7* for the gut–brain axis (49). In addition to the aforementioned genes, *RPL3* is the one present in centers of healthy controls and pediatric/adult CD networks. *RPL3* is marked by rs12627970, and the susceptibility gene related to this locus was annotated as *SYNGR1* in previous studies (32). In Europeans, the G allele of rs12627970 was reported (6) to increase the risk of IBD by an odds ratio of 1.12 ($P = 1.94 \times 10^{-18}$). Marigorta et al showed that rs12627970 is an eQTL in human blood (32). Using the webtools of GTEx (50), we discovered that the eQTL effect of rs12627970 in the small intestine (terminal ileum) is significantly associated with *SYNGR1* ($P = 4.99 \times 10^{-10}$) and *RPL3* ($P = 4.6 \times 10^{-3}$). However, the gnomAD pLI score (the probability of being LoF-intolerant) showed a possibility of 0.45 for *SYNGR1* in a LoF gene, which is smaller than 0.99 of *RPL3*. The expression of *SYNGR1* in monocytes is infrequent, with less than 0.1 proportion of cells expressing it. Therefore, it was excluded from the network. Its strictness measure is 0.14, which is smaller than 3.23 of *RPL3*, indicating the dosage of *RPL3* in monocytes is more intolerant to be altered by eQTL than *SYNGR1*. *RPL3* encoded ribosomes that catalyze

the protein synthesis, with its function implicated in a number of biological processes. Numerous pieces of evidence indicated a subset of ribosomal proteins regulate the cell cycle and apoptosis (51). Moreover, *RPL3* interacts with *DUOX2* that promotes the progression of colorectal cancer cells (52). In colorectal cancer cells, knockdown of *DUOX2* inhibits invasion and migration that can be reversed by the overexpression of *RPL3*. The loss of *RPL3* plays an important role in inhibition of cell proliferation upon exposure to actinomycin D (a widely used anticancer drug) (53). In our study, *RPL3* was indicated as a key regulator at the network center. Collectively, the *RPL3* gene may be a regulatory target that is worth an attention.

In GWAS that hypothesized SNPs are independent (10), an interaction set of SNP × SNP in the regression model can be used to investigate the gene–gene interactions. However, the power of detecting significant SNP–SNP interactions is constrained by GWAS sample size. Moreover, *n* SNPs in a large sample size request significant computational performance necessary for calculating $n \times (n - 1)/2$ pairwise SNP–SNP interactions. The requirement of sample size and computational performance make the GWAS challenge to discuss gene–gene interaction. A gene network based on co-expression, protein–protein interaction, and/or functional annotations such as GO/KEGG can shed light on the study of gene–gene interactions. Benefiting from the human cellar landscape, we propose the GGM for the gene expression network at the single-cell level using single-cell transcriptome data. Firstly, we identify the gene-specific expression

cell categories by the EWCE method. Subsequently, we show the gene expression dosage characteristic in the cell type by the strictness method. Finally, we build a network graph on the gene expression in single-cell population by the GGM. This strategy may be widely applied in post-GWAS and post-scRNA studies for investigating the involvements of disease-related cells, the dosage requirement of disease-related genes, and the role of gene–gene interactions.

In conclusion, our analysis revealed that most of the IBD susceptibility genes are specifically and strictly expressed in monocytes. The susceptibility gene connections generate an expression network, which is robust for expression balance to prevent genetic or environmental alteration. In contrast, the network is disconnected in CD patients, suggesting the gene network contributed to the CD pathogenesis. These findings provide novel insight into the IBD etiology.

# Materials and Methods

## IBD susceptibility genes

We employed 232 IBD risk loci reported by the largest cohort GWAS (86,640 European participants and 9,846 participants of East Asian, Indian, or Iranian descent) (6). The variant effect predictor tool (54) was used to annotate the genes located within or close to the loci. A total of 232 genes were identified as IBD susceptibility genes and used for further analysis (Table S1).

## IBD susceptibility genes with eQTL

We employed 104 genes with eQTL associated with IBD risk loci reported by Momozawa et al (29) (Table S1). A total of 43 genes were overlapped with the 232 IBD susceptibility genes within or near to the IBD risk loci. We accessed the coding variants at these 43 genes from the gnomAD database and annotated the variant consequence by the variant effect predictor. We classified the start lost, stop gained, stop lost, splice acceptor/donor variant, and frameshift into the LoF variant. We calculated the CAF of $n$ LoF variants with allele frequency (AF) < 0.05 in the gnomAD database by the following formula:

$$CAF = 1 - \prod_{i=1}^{n}(1 - AF_i)$$

## Haploinsufficient genes and LoF-tolerant genes

We employed 299 haploinsufficient (HI) genes predicted by Dang et al (55) and 330 putative homozygous LoFT genes predicted by Lek et al (30) (Table S1). To the comparison of the dosage characteristic between IBD susceptibility genes and known genes, we used HI genes as a positive control of dosage-sensitive genes and LoFT genes as negative controls.

## Single-cell transcriptome datasets

In the discovery phase, we employed a large single-cell transcriptome dataset of the human intestinal tract from Elmentaite et al's study (21). The data include the expression of 33,538 genes in 428,000 high-quality cells from up to five anatomical regions in the developing and up to 11 distinct anatomical regions of the pediatric and adult human intestinal tract. These cells were classified into six main cell types, including epithelial/mesenchymal/endothelial/immune/neural/erythroid, and 133 subtypes/states. In the replication phase, we employed an independent single-cell transcriptome dataset from Kong et al's study (25) to validate the main results. The data include the expression of 27,345 genes in 720,633 high-quality cells from the terminal ileum and colon of 46 CD patients and 25 healthy adults. These cells were classified into three main cell types, including immune/epithelial/stromal, and 54 subtypes/states.

## Expression weighted cell-type enrichment

A method named EWCE (56) (https://github.com/NathanSkene/EWCE) was used to examine the expression specificity across multiple cell types from single-cell transcriptomes. Initially, we calculated the specificity of genes in each cell type using the generate-celltype-data function. We then used the bootstrap-enrichment-test function to estimate the $P$-value of target gene specificity. The bootstrap approach randomly samples 10,000 gene lists containing the same number of target genes. The specificity of these 10,000 gene lists served as the distribution background. The cumulative density function of the specificity distribution and the FDR method were used to calculate the $P$-values of the specificity of target genes. A threshold of FDR $P$-value < 0.05 is used to indicate the cell type that enriched the expression of disease-related genes.

## Dosage requirement in related cells

In our previous study (18), the strictness measure was defined to estimate the dosage requirement for a given gene. Strictness was calculated by the SD of fold change: $S = 1/\sqrt{\frac{\sum_{i}^{n}(C_i - \overline{C})^2}{n-1}}$, in which $S$ denotes strictness, $C$ refers to fold change, $i$ refers to the $i$th cell, and $n$ refers to the total number of cells. The fold change was determined by dividing the expression of a single cell by the mean expression of all cells: $C_i = E_i/\overline{E}$, where $E$ refers to the expression in a single cell, $\overline{E}$ refers to the mean expression, and $i$ refers to the $i$th cell. A high strictness value indicates that a gene's expression must be strict. A low strictness indicates that the gene expression is tolerant to alterations.

We compared the dosage characteristics of IBD susceptibility genes with those of the HI and LoFT gene sets via a rank sum test. To calculate the significance of strictness for target genes relative to background genes, we estimated the normal distribution of average strictness using the central limit theorem. Firstly, we identify the number ($n$) of target genes that we wish to examine, and then calculate the strictness mean ($x$) for these $n$ genes. Subsequently, we randomly select $n$ genes from all the genes and repeat the sampling 500,000 times via the bootstrap method. Finally, we calculate the strictness means for each of the 500,000 random samples and then use the maximum-likelihood estimation to estimate the distribution's mean ($\mu$) and the SD ($\sigma$). The distribution of sample means should approximate the normal distribution: $X \sim N$

$(\mu, \sigma^2)$ . $P(x > X) = 1 - \frac{1}{\sigma\sqrt{2\pi}} \int_{-\infty}^{X} \exp\left\{-\frac{(x-\mu)^2}{2\sigma^2}\right\} dx$ is the formula used to determine the $P$-value of the mean of the target genes.

### Network analysis

We firstly used Pearson's correlation coefficient (r) to estimate the connection between two genes from their expression. As we know there is a network among genes, the correlation coefficients are not independent, which means a connection with a correlation coefficient may be a marginal effect caused by another strong connection. Subsequently, a partial correlation coefficient ($\rho$) was used to adjust a correlation coefficient by other genes. The GGM was used to construct the gene expression network (57). We define a network as $G = (V,E)$, where $V$ refers to the nodes, and $E$ refers to the edge connecting two nodes. The GGM defined the following:

$$E = \left\{\{i,j\} \in V : \rho_{ij} | V\{i,j\} \neq 0\right\}$$

$$\rho_{ij} | V\{i,j\} = -\omega_{ij} \Big/ \sqrt{\omega_{ij}\omega_{ij}}$$

$$\left(\omega_{ij}\right)_{n \times n} = \begin{pmatrix} c_{11} & \cdots & c_{1n} \\ \vdots & \ddots & \vdots \\ c_{n1} & \cdots & c_{nn} \end{pmatrix}^{-1}$$

$$c_{ij} = E[X_i - E(X_i)]\left[X_j - E(X_j)\right]$$

$$X = (X_1, X_2, X_3, ..., X_n)^T$$

where $X$ refers to the expression matrix of $n$ genes in a single cell, which is calculated from read number $C$: $X = \log(1 + C)$. Genes with a proportion of read counts in all cells less than 0.9 were excluded. The huge R package was used to construct the graph $G = (V, E)$. The ggnetwork R package was used to visualize the network graph. The graph central score $c(v)$, graph density $d(G)$, and graph clustering coefficient (58) $cl(v)$ were calculated by

$$c(v) = \sum_{s \neq t \neq v \in V} \frac{\sigma(s,t|v)}{\sigma(s,t)}$$

where $\sigma(s,t|v)$ refers to the min counts of edges linked to node $s$ and node $t$ through node $v$, and $\sigma(s,t)$ refers to the min counts of edges linked to node $s$ and node $t$.

$$d(G) = \frac{E}{V(V-1)/2}$$

where $E$ refers to the edges, and $V$ refers to the nodes.

$$cl(v) = \frac{\left(A + A^T\right)^3_{vv}}{2\left[d_v\left(d_v - 1\right) - 2\left(A^2\right)_{vv}\right]}$$

where $A$ refers to the adjacency matrix, and $d$ refers to the degree.

To estimate the proportion of true gene–gene connections, we simply employed the protein–protein interactions of the STRING (31) database (v11.5; https://cn.string-db.org) as a truth set. The STRING database identified a number of 11,938,498 protein–protein interactions from a space of sample of $\frac{n(n-1)}{2}$ interactions, where $n$ refers to the number of 19,566 protein-coding genes. The proportion of truth interactions from random sampling is 0.062. A binomial test was used to indicate the significance of true gene–gene connections from the connections identified by the GGM.

## Data and Code Availability

IBD susceptibility locus, susceptibility genes, susceptibility genes with eQTL, HI genes, and LoFT genes are listed in Table S1. The code for the study is written in the R program and released on GitHub (https://github.com/liuhankui/IBD).

### Ethics statement

This study was reviewed and approved by the BGI-Shenzhen Ethics Review Committee. There were no participants or donors involved in our research.

## Supplementary Information

## Acknowledgements

We would like to thank the providers of the public databases and software we used in our study. This study was supported by the National Key Research and Development Program of China (No. 2022YFC2703102), Hebei Industrial Technology Research Institute Construction Project (No. 235790429H), and Shenzhen Science and Technology Program (No. KCXFZ20201221173208023).

### Author Contributions

H Liu: conceptualization, data curation, software, formal analysis, investigation, visualization, methodology, and writing—original draft, review, and editing.
L Guan: data curation, formal analysis, and writing—review and editing.
X Su: formal analysis and writing—review and editing.
L Zhao: resources, supervision, project administration, and writing—review and editing.
Q Shu: resources, supervision, funding acquisition, validation, investigation, and writing—review and editing.
J Zhang: conceptualization, resources, supervision, funding acquisition, investigation, and writing—review and editing.

### Conflict of Interest Statement

The authors declare that they have no conflict of interest.

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
