## [Reviewer comments · Life Science Alliance]

Life Science Alliance

A broken network of susceptibility genes in the monocytes of Crohn's disease patients

Hankui Liu, Liping Guan, Xi Su, Lijian Zhao, Qing Shu, and Jianguo Zhang

DOI: <https://doi.org/10.26508/lsa.202302394>

Corresponding author(s): Hankui Liu, BGI Group

Review Timeline:

Submission Date:	2023-09-26
Editorial Decision:	2023-11-20
Revision Received:	2024-04-09
Editorial Decision:	2024-05-21
Revision Received:	2024-06-15
Accepted:	2024-06-17

Transaction Report:

November 20, 2023

Re: Life Science Alliance manuscript #LSA-2023-02394-T

Dr. Hankui Liu
BGI Group
Yantian road, BGI
Shenzhen 518000
China

Dear Dr. Liu,

Thank you for submitting your manuscript entitled "A broken network of IBD-susceptibility genes in the monocytes of IBD patients". The manuscript has been evaluated by expert reviewers, whose reports are appended below. Unfortunately, after an assessment of the reviewer feedback, our editorial decision is against publication in Life Science Alliance.

Although your manuscript is intriguing, I feel that the points raised by the reviewers are more substantial than can be addressed in a typical revision period. If you wish to expedite publication of the current data, it may be best to pursue publication at another journal.

Given the interest in the topic, I would be open to re-submission to Life Science Alliance of a significantly revised and extended manuscript that fully addresses the reviewers' concerns and is subject to further peer review. If you would like to resubmit this work to Life Science Alliance, you may submit an appeal directly through our manuscript submission system. Please note that priority and novelty would be reassessed at re-submission.

Regardless of how you choose to proceed, we hope that the comments below will prove constructive as your work progresses.

Thank you for thinking of Life Science Alliance as an appropriate place to publish your work.

Sincerely,

Reviewer #1 (Comments to the Authors (Required)):

The manuscript by Liu et al contains a computational analysis of data available in various publically available data bases containing information about IBD susceptibility genes. The authors point out that the 232 IBD-susceptibility gene loci previously identified account for only about 25% of the heritability of IBD and that since these loci are independent (i.e., their increase association with IBD does not depend on any other locus) the missing heritability may be due to defects in gene interaction. Previous studies to examine this possibility via network analysis were limited by the fact that they were reliant on data derived from bulk cell populations. In this study the authors utilize more recent data derived from single cell mRNA studies to perform more discerning network analysis of IBD susceptibility genes. The authors main conclusion is that in IBD this network is "disconnected."

Specific Comments:

1. The data provided in Figure 1A is poorly annotated in the figure legend since the latter does not clearly indicate that the listed cells are ranked according to their enrichment of IBD genes. In any case, these data may be somewhat misleading if it is based on data derived from pediatric IBD which is skewed towards Crohn's disease rather than ulcerative colitis, the latter a disease more likely to be associated with epithelial cell abnormalities. In addition, the prominence of ILC3 cells may also relate to the use of data derived from children since this cell may contribute to early rather than persistent disease.
2. The finding depicted in Figure 1A showing that in diseased IBD tissue the cells most enriched in expression of disease susceptibility genes are monocytes and Th1 cells is hardly surprising since it is already known that these gene affect immune responses. Assuming the data relate mostly to Crohn's disease what is somewhat new is that epithelial cells are not a category of cells expressing susceptibility genes given the great number of studies focusing on the relation of autophagy and cell stress

responses (involving XBP-1 polymorphisms).to IBD. This deserves comment.

3. It is absolutely essential that the data shown in Figure 1A be derived by identification of cells by a unique set of positive (and negative) cell markers given the overlap in gene expression (or non-expression). ILC3 cells, for instance, express genes also expressed by Th17 cells (or even Th1 cells). This requirement needs be more clearly affirmed.

4. The authors indicate that based on prior studies of ALS LoF genes correlates with high "strictness" whereas GoF correlates with low "strictness". However, they provide no evidence that this is necessarily so in IBD or any other disease. In addition, they state that the high strictness indicate the genes are "dosage sensitive." This does not necessarily follow and. Requires verification.

5. The main finding in this study is that by network analysis key central genes are "disconnected" from other genes and therefore many genes are independent and not under network regulation. A SNP in one of these central genes, RPL3, is suggested to be an important polymorphism and therefore a therapeutic target despite having a low odds ratio (1.12). It seems unlikely this would be an IBD-specific disease gene since it encodes a protein important to several basic molecular functions possible involved in proliferative process necessary for inflammation.

6. A basic assumption of this computational analysis is that the abnormal network is a cause rather than an effect of the inflammation. It seems possible, or even likely, that a severe inflammatory process of any kind would cause a disconnected network; the authors need to provide data focused on networks in pediatric patients with IBD who are in complete remission to address this question.

Reviewer #2 (Comments to the Authors (Required)):

It is an ongoing challenge to maximum utilize the huge sequencing data generated so far to explore and understand the pathogenesis of diseases and identification of potential therapeutic targets. The authors have leveraged the use of genetic and single-cell RNA sequencing analysis to address the IBD-susceptibility genes that are identified by Genome-wide association studies. The authors used mathematic and bioinformatic methods to investigate the IBD- susceptibility gene network. This approach may be used for other diseases as well to find out the most involved cell types and gene networks. Their analysis shows that the human gut monocytes express most of these genes followed by ILC3 cells and TH1 cells. This is in line with the current knowledge regarding these cell types in this field. Even though this is a different approach to show the involvement of these cell types in IBD, the finding cannot be considered as novel. They also show that these genes are dosage-sensitive in monocytes. Other than the strategy used by the authors, the difference in IBD- susceptibility genes in IBD patients and healthy individuals is the only interesting and novel finding of the paper. They show the robustness of IBD-susceptibility gene network in healthy monocytes and the gene network get disconnected in in IBD patients.

Comments:

Introduction part: The authors have indicated the gene network construction of Jimmy et al, Mesbah-Uddin et al, Lauren et al. Please also mention the major findings of these papers. For example, the genes or cells they identified using their approach. This will help to indicate the need and significance of gene network-based analysis related studies in diseases.

Introduction part: Please include more background and relevance regarding the significance of studying and addressing the role of IBD- susceptibility genes and IBD related cell types, GWAS and GGM.

Methods section "IBD-susceptibility genes with eQTL": Please mention the full form of "eQTL" and "VEP".

Methods section "Single-cell transcriptome dataset": Please mention the unique identifiable number of the sc-RNA data set.

Methods section "Network analysis": The authors mentioned that "Genes with a proportion of read counts in all cells less than 0.9 were excluded". It would be useful to specify how this threshold was selected. Could this result in the loss of any important genes?

Results section "Cells enriched the expression of IBD-susceptibility genes": The authors mentioned "The involvements of monocytes, ILC3 cells and TH1 cells revealed in our study were consistent with previous reports". Please add reference to the previous reports. And it would be good to give a brief note about these previous reports in the introduction part.

Results section "Cells enriched the expression of IBD-susceptibility genes": It would be good to include a brief overview/conclusion regarding the other cell types in this analysis.

Results section "IBD-susceptibility genes are dosage-sensitive in monocytes": Is it possible to check the expression network of IBD- susceptibility genes of in Th1 cells since they were also increased in IBD patients compared to healthy?

Results section "IBD-susceptibility genes are dosage-sensitive in monocytes": Please add reference for the claim "Our prior research demonstrated that ALSrisk genes with known mechanism of LoF exhibit high strictness and the gain of function genes exhibits low strictness".

Results section "Connections of IBD-susceptibility genes are broken in IBD patients": The authors mentioned that "Using a threshold of degree >5 and central score > 180 , we identified 2, 5, 8 central genes in the fetal, paediatric, and adult healthy networks, respectively.". How were these thresholds chosen and what is its significance?

Results section "Connections of IBD-susceptibility genes are broken in IBD patients": Performing pathway analysis (GO/KEGG) using the highly connected genes will give an idea about the difference in pathways that are enriched in IBD patients and healthy individuals.

Results section "Connections of IBD-susceptibility genes are broken in IBD patients": If possible, the authors can interpret the results in a more deep and critical manner. Here other than the central gene and a few other genes of the network, it seems like the other genes are ignored. Exploring and studying about the genes that are a part of broken network could also give us some idea about the etiology of IBD. Maybe it would be helpful if the authors could mention some already known information from literature regarding some of the interesting genes that are part of the broken network.

Discussion part: This part needs to be written in a more focussed manner.

The reader might not have a continuation while reading sentences like this: "GWAS hypothesized that the loci are independent. Polygenic score method hypothesized that the locus' effect on phenotype is additive. SNP-SNP interaction GWAS can investigate the gene-gene interactions, but its efficacy is constrained by sample size and the computational performance necessary for calculating $n \times (n - 2)/2$ pairwise SNP-SNP interactions for n SNPs. A gene network based on co-expression, protein-protein interaction, and/or functional annotations such as GO/KEGG can shed light on gene-gene interactions".

Discussion part: The sentences like this should be rephrased and re-written: "We calculated the cumulative allele frequency (CAF) of low frequency (allele frequency < 0.05) LoF mutation of 43 IBD-susceptibility genes with eQTL (Supplementary Table 1) in gnomAD database and showed a LoF CAF of 0.1, indicating that an individual has a possibility of 0.1 to carry at least a LoF allele at the 43 IBD-susceptibility genes. Why there can be such a common LoF CAF in a population. The gene expression network in the healthy adults and paediatric can be a explain".

Discussion part: It would be helpful to give a brief overview of what is already known about RPL3 to indicate the novelty of the authors findings.

Discussion part: Even though the authors reveal genes such as STAT3, ZFP36L1, and TET2, along with RPL3 as central regulators in health, they do not mention what is the biological meaning of this finding or its relevance.

Reviewer #3 (Comments to the Authors (Required)):

Liu et al 2023 - Life Science Alliance

In this study the authors set out to examine the expression of key IBD susceptibility gene signatures at single cell level using publicly available data. In particular, they generate a catalogue of IBD susceptibility loci candidates from a number of studies and examine expression of this within one of the prominent cell atlases of human (both adult and paediatric) intestine. The authors conclude that the 'majority' of IBD susceptibility loci are found in monocytes in IBD. As detailed below, although the manuscript is easy to follow and the data presented in a clear and concise manner, there are major concerns about the approach used by the authors which very much undermines the conclusions reached.

General and major comments

1. The main concern here, is the imputed public datasets that have been used. The Elmentaite 2021 study by the Teichmann group was one of the first cell atlases of human intestine and has a large number of cells 428,000. However, these are tremendously variable, which introduces significant confounding variable in the analysis. For example, tissue site (11 distinct locations in the adult samples) and developmental stage (e.g. foetal/paediatric and adult). Most importantly, there are only 7 IBD samples analysed, all of which derive from paediatric samples. How much phenotypic information is known on the paediatric samples? E.g. age of onset, are these VEOIBD, which is a discrete entity, etc. Similarly, what is the ethnicity of the different datasets used in this study, particularly the paediatric patients as we know ethnicity will affect IBD susceptibility traits.
2. For this study to draw the conclusions it currently has, analysis would need to be done across multiple studies to demonstrate the validity of the authors' conclusions that 'majority of IBD susceptibility loci are specifically expressed in monocytes'. For example, the Kong et al. study by the Xavier group in which 46 Crohn's disease samples were compared with 25 healthy controls would be an ideal resource with which to perform this analysis (PMID: 36720220). This is important because paediatric

onset IBD represents between 10-25% of all IBD, and the genetic susceptibility underpinning this is likely to be subtly different to adult onset IBD (e.g. PMID: 22543157). As a result, using a small number of paediatric IBD samples matched to adult GWAS data is not the most appropriate paired analysis to be performed.

Specific comments

1. It is unclear how the graph in Figure 1A supports the conclusion that the 'majority of IBD susceptibility loci are specifically expressed in monocytes'. As the authors state themselves, this enrichment analysis equally shows that the IBD gene signature is expressed by Th1 cells and ILC3. Data supporting the author's conclusion is not shown in this figure. For instance, what proportion of the 232 loci were enriched in each of the cell types presented in Fig 1A? How was this weighted, e.g. is the variance explained by a small number of SNPs?
3. The data presented in Figure 1B are unclear. What exactly is being shown here? Proportion of ILC3, Th1 cells and monocytes of the entire dataset? Is this meaningful? The authors talk about cell numbers in the text but only shown proportion data. Given monocytes are the focus for analysis, the absolute numbers of these cells in each dataset, in each location and developmental stage, should be presented. Have the monocytes from multiple anatomical locations been pooled for example?
4. The rationale for using the current approach is not clear, specifically, why would IBD loci necessarily be expected to act in a network? Would it have been better to look at networks that were different between groups and overlay loci to this?
5. The claim that "The density and cluster of IBD-susceptibility genes network is lower in paediatric IBD patients than in healthy participants" could simply reflect the low power ($n=7$) of these data.

Minor comments

1. There are a few controversial statements in the introduction that required explanation / citation e.g. "Genetics is the primary driver of IBD with an estimated heritability of 0.75 (CD) and 0.67 (UC) from twin cohort". The authors cite one study to support this statement. The changes in IBD incidence alone in newly-industrialised countries shows that genetics alone cannot be responsible as the primary driver of IBD
2. "The majority of IBD risk loci share consistent effect in CD and UC, and also in European and non-European populations". But these will also be different in many important areas, e.g. IL23 versus NOD2
3. "As the GWAS strategy hypothesized that loci are independent, the absent heritability could be due to gene interactions". But could also be due to other factors, e.g. LD, epigenetics etc.
4. "These loci explained 13.1% and 8.2% of the disease susceptibility variance for CD and UC, respectively". Citation required

Appeal Request

Dear Editors,

Thank you for the opportunity to resubmit a revision of this manuscript (#LSA-2023-02394-T). We would also like to take this opportunity to express our thanks to the reviewers for the positive feedback and helpful comments for correction or modification. We believe have revised an improved manuscript to address the reviewer comments, which you will find uploaded alongside this letter. Briefly, we validated the main results in an independent single-cell transcriptome dataset from the colon of 46 CD patients and 25 healthy adults. The R code for discovery and replication is publicly available on GitHub (<https://github.com/liuhankui/IBD>) with detailed step-by-step script. We anticipate that the revised manuscript meets the rigorous standards of Life Science Alliance.

Sincerely yours,

Hankui
Email: liuhankui@genomics.cn

Decision Letter for Appeal

December 7, 2023

MS: LSA-2023-02394-T

Dr. Hankui Liu
BGI Group
Yantian road, BGI
Shenzhen 518000
China

Dear Dr. Liu,

Your manuscript entitled "A broken network of IBD-susceptibility genes in the monocytes of IBD patients" has now been reconsidered, and I am pleased to let you know that we have decided to send your manuscript back to the original Referees for external review.

Please use the following link to submit your manuscript:

<https://lsa.msubmit.net/cgi-bin/main.plex?el=A7Na6BGA7A7CqwE2I4B9ftdjhma1fUsJEXwl1324oi7QZ>

Yours sincerely,

Eric Sawey, PhD
Executive Editor
Life Science Alliance
<http://www.lsa-journal.org>

Reviewer #1 (Comments to the Authors (Required)):

The manuscript by Liu et al contains a computational analysis of data available in various publically available data bases containing information about IBD susceptibility genes. The authors point out that the 232 IBD-susceptibility gene loci previously identified account for only about 25% of the heritability of IBD and that since these loci are independent (i.e., their increase association with IBD does not depend on any other locus) the missing heritability may be due to defects in gene interaction. Previous studies to examine this possibility via network analysis were limited by the fact that they were reliant on data derived from bulk cell populations. In this study the authors utilize more recent data derived from single cell mRNA studies to perform more discerning network analysis of IBD susceptibility genes. The authors main conclusion is that in IBD this network is "disconnected."

Specific Comments:

1. The data provided in Figure 1A is poorly annotated in the figure legend since the latter does not clearly indicate that the listed cells are ranked according to their enrichment of IBD genes. In any case, these data may be somewhat misleading if it is based on data derived from pediatric IBD which is skewed towards Crohn's disease rather than ulcerative colitis, the latter a disease more likely to be associated with epithelial cell abnormalities. In addition, the prominence of ILC3 cells may also relate to the use of data derived from children since this cell may contribute to early rather than persistent disease.

Response: Thanks for your critical comment. The figure legends have been revised to specify that the mentioned cells are ranked according to their enrichment of IBD genes. The scRNA data used in our study pertains to pediatric patients with Crohn's disease (CD), as reported in the paper. In the amended paper and figures, we have replaced the term "pediatric IBD" with "pediatric CD". In addition, we added a table (table 1) that presents a concise overview of the scRNA dataset. Finally, we corroborate our findings using a replication study that utilized the scRNA of 46 adult CD patients and 25 healthy adults. [methods: 114-118; results: 200-207, 234-236,

242-246, 255-267]. The association of ILC3 in pediatric CD did not validate in adult CD, this result is consistent with your comment [line: 205-207].

2. The finding depicted in Figure 1A showing that in diseased IBD tissue the cells most enriched in expression of disease susceptibility genes are monocytes and Th1 cells is hardly surprising since it is already known that these genes affect immune responses. Assuming the data relate mostly to Crohn's disease what is somewhat new is that epithelial cells are not a category of cells expressing susceptibility genes given the great number of studies focusing on the relation of autophagy and cell stress responses (involving XBP-1 polymorphisms) to IBD. This deserves comment.

Response: Using the term “Interestingly” in the statement “Interestingly, the number of monocytes was obviously (Fisher’s exact test P -value = 2.7×10^{-108}) increased in patients with IBD” may not be advisable. This result is intriguing to bioinformaticians, but not to researchers in the field of IBD. This sentence has been altered [line:188-193].

3. It is absolutely essential that the data shown in Figure 1A be derived by identification of cells by a unique set of positive (and negative) cell markers given the overlap in gene expression (or non-expression). ILC3 cells, for instance, express genes also expressed by Th17 cells (or even Th1 cells). This requirement needs to be more clearly affirmed.

Response: Affirmative. ILC3 cells are the innate counterparts of Th17 cells³⁴ [line: 197]. There is a possibility of gene expression overlapping between different cell subtypes. We employed the cell types identified in the study, together with the corresponding read counts of the cells. Th17 cells exhibit a suggestive significance (FDR is closed to 0.05) (Fig. 1A) in pediatric CD and significance (FDR < 0.05) in adult CD (Fig. S1) [line: 203-205]. Due to the study’s focus on network outcomes, we did not prioritize the cell type with suggestive relevance.

4. The authors indicate that based on prior studies of ALS LoF genes correlates with high "strictness" whereas GoF correlates with low "strictness". However, they provide no evidence that this is necessarily so in IBD or any other disease. In addition, they state that the high strictness indicate the genes are "dosage sensitive." This does not necessarily follow and. Requires verification.

Response: We regret the lack of precision in our previous dosage sensitivity finding. We verified the outcome of "dosage sensitive" by employing the Z-score of the expected/observed ratio of loss of function mutation recorded in gnomAD (Fig 1E). The Z-score method was used to estimate the tolerance or intolerance of a gene against loss of function mutation¹. In relation to the indispensability of a strictness analysis in IBD genes, we perform the strictness analysis for two specific objectives. Firstly, we employed rigorous criteria to show the distinctive features of IBD genes. These genes are strictly required in monocytes, meaning that any significant change in their expression, such as a loss of function mutation, will induce damaging effect to the gene/cell function. Subsequently, we showed that the IBD susceptibility genes harbor a common cumulative allelic frequency of loss of function mutations, as recorded in the gnomAD database. This implies the existence of a network of gene-gene interactions that govern the genes affected by loss of function mutations.

5. The main finding in this study is that by network analysis key central genes are "disconnected" from other genes and therefore many genes are independent and not under network regulation. A SNP in one of these central genes, *RPL3*, is suggested to be an important polymorphism and therefore a therapeutic target despite having a low odds ratio (1.12). It seems unlikely this would be an IBD-specific disease gene since it encodes a protein important to several basic molecular functions possible involved in proliferative process necessary for inflammation.

Response: We apologize for the lack of clarity in our discussion of the role of *RPL3*. *RPL3* encoded ribosomes that regulate the cell cycle and apoptosis². While *RPL3* is involved in the basic molecular functions, the odds ratio (1.12) of *RPL3* loci indicates that the mutation of *RPL3* observed in IBD did not induce large damaging to its normal function. Furthermore, it has been revealed that *RPL3* plays a role in colorectal cancer. *RPL3* interacts with *DUOX2* that promotes the progression of colorectal cancer cells³. In colorectal cancer cells, knockdown of *DUOX2* inhibits invasion and migration that can be reversed by the overexpression of *RPL3*. The loss of *RPL3* plays an important role in inhibition of cell proliferation upon exposure to Actinomycin D (a widely used anti-cancer drug)⁴. *RPL3* is indicated as a key gene in the regular center of network. Collectively, *RPL3* gene may be a therapeutic target that worth for an attention. [line: 325-347]

6. A basic assumption of this computational analysis is that the abnormal network is a cause rather than an effect of the inflammation. It seems possible, or even likely, that a severe inflammatory process of any kind would cause a disconnected network; the authors need to provide data focused on networks in pediatric patients with IBD who are in complete remission to address this question.

Response: Exemplary comment. In order to answer this question, we employed a supplementary dataset consisting of 46 adult CD patients and 25 healthy controls. We construct three networks of susceptibility genes using scRNA data obtained from healthy colon of healthy adults, non-inflamed colon of adult CD, and inflamed colon of adult CD, respectively. We showed that the density of network of non-inflamed colon is significantly lower than that of healthy colon. This finding is included in the result [line: 255-267]. Thank you for your nice comment.

Reviewer #2 (Comments to the Authors (Required)):

It is an ongoing challenge to maximum utilize the huge sequencing data generated so far to explore and understand the pathogenesis of diseases and identification of potential therapeutic targets. The authors have leveraged the use of genetic and single-cell RNA sequencing analysis to address the IBD-susceptibility genes that are identified by Genome-wide association studies. The authors used mathematic and bioinformatic methods to investigate the IBD- susceptibility gene network. This approach may be used for other diseases as well to find out the most involved cell types and gene networks. Their analysis shows that the human gut monocytes express most of these genes followed by ILC3 cells and TH1 cells. This is in line with the current knowledge regarding these cell types in this field. Even though this is a different approach to show the involvement of these cell types in IBD, the finding cannot be considered as novel. They also show that these genes are dosage-sensitive in monocytes. Other than the strategy used by the authors, the difference in IBD-susceptibility genes in IBD patients and healthy individuals is the only interesting and novel finding of the paper. They show the robustness of IBD-susceptibility gene network in healthy monocytes and the gene network get disconnected in in IBD patients.

Comments:

Introduction part: The authors have indicated the gene network construction of Jimmy et al, Mesbah-Uddin et al, Lauren et al. Please also mention the major findings of these papers. For example, the genes or cells they identified using their approach. This will help to indicate the need and significance of gene network-based analysis related studies in diseases.

Response: Thanks for your suggestion. We have included a concise summary of the findings from these investigations [line: 59-65].

Introduction part: Please include more background and relevance regarding the significance of studying and addressing the role of IBD- susceptibility genes and IBD related cell types, GWAS and GGM.

Response: I appreciate your suggestion. We added a brief of IBD- susceptibility genes and IBD related cell types in background [line: 73-77]. GWAS were introduced in the initial section of the background [line: 46-53]. The use of GGM was mentioned in the methodology section.

Methods section "IBD-susceptibility genes with eQTL": Please mention the full form of "eQTL" and "VEP".

Response: We have added the full form of "eQTL" and "VEP" in revised manuscript. [line:92, 88]

Methods section "Single-cell transcriptome dataset": Please mention the unique identifiable number of the sc-RNA data set.

Response: We have added the website for the sc-RNA data. [Table 1]

Methods section "Network analysis": The authors mentioned that "Genes with a proportion of read counts in all cells less than 0.9 were excluded". It would be useful to specify how this threshold was selected. Could this result in the loss of any important genes?

Response: The threshold was estimated from the proportion distribution where largely differ the left and right distribution. At the first part of result, we identified the IBD-related cell type that enriched the expression of IBD genes. If a gene is

infrequently expressed in the specific cell type, it is likely not a significant gene for the cellular function associated with the disease.

Results section "Cells enriched the expression of IBD-susceptibility genes": The authors mentioned "The involvements of monocytes, ILC3 cells and TH1 cells revealed in our study were consistent with previous reports". Please add reference to the previous reports. And it would be good to give a brief note about these previous reports in the introduction part.

Results section "Cells enriched the expression of IBD-susceptibility genes": It would be good to include a brief overview/conclusion regarding the other cell types in this analysis.

Response: We have added the citations in revised manuscript and a summary of IBD-related cells type to introduction section [line: 73-77]. We have also added a concise description of the cell types documented in the previous study [line: 108-118].

Results section "IBD-susceptibility genes are dosage-sensitive in monocytes": Is it possible to check the expression network of IBD- susceptibility genes of in Th1 cells since they were also increased in IBD patients compared to healthy?

Response: Appreciative remark. We are also interested in examining the network status of Th1 cells in patients with IBD and comparing it to that of healthy individuals. Nevertheless, the number of Th1 cells is 0 in healthy fetal individuals, 31 in healthy paediatric individuals, and 4 in healthy adult individuals. The cell number 31 is insufficient for calculating the covariance matrix for network analysis (Fig. 1B).

Results section "IBD-susceptibility genes are dosage-sensitive in monocytes": Please add reference for the claim "Our prior research demonstrated that ALSrisk genes with known mechanism of LoF exhibit high strictness and the gain of function genes exhibits low strictness".

Response: The citations have been added into the revised manuscript.

Results section "Connections of IBD-susceptibility genes are broken in IBD patients": The authors mentioned that "Using a threshold of degree >5 and central score > 180 , we identified 2, 5, 8 central genes in the fetal, paediatric, and adult healthy networks, respectively.". How were these thresholds chosen and what is its significance?

Response: The threshold of network center is differed by the density of whole graph. We determined these thresholds by the joint distribution of degree and central score to indicate the genes with the highest number of interactions.

Results section "Connections of IBD-susceptibility genes are broken in IBD patients": Performing pathway analysis (GO/KEGG) using the highly connected genes will give an idea about the difference in pathways that are enriched in IBD patients and healthy individuals.

Results section "Connections of IBD-susceptibility genes are broken in IBD patients": If possible, the authors can interpret the results in a more deep and critical manner. Here other than the central gene and a few other genes of the network, it seems like the other genes are ignored. Exploring and studying about the genes that are a part of broken network could also give us some idea about the etiology of IBD. Maybe it would be helpful if the authors could mention some already known information from literature regarding some of the interesting genes that are part of the broken network.

Response: Nice comment. We perform a KEGG pathway enrichment analysis and demonstrated the presence of six pathways with a false discovery rate (FDR) P -value less than 0.01 (**Fig. S3**). Jak-STAT signaling pathway⁵, HIF-1 signaling pathway⁶, T_h17 cell differentiation⁷ were known to play important role in IBD. Over the last decade, programmed death ligand 1 (PD-L1) expression were proposed as a key mechanism for the mucosal tolerance in the gut⁸. Recently study suggested that Chronic *Toxoplasma gondii* infection enhances monocyte activation to increase inflammation associated with a secondary environmental insult⁹.

Discussion part: This part needs to be written in a more focussed manner.

The reader might not have a continuation while reading sentences like this: "GWAS hypothesized that the loci are independent. Polygenic score method hypothesized that the locus' effect on phenotype is additive. SNP-SNP interaction GWAS can investigate the gene-gene interactions, but its efficacy is constrained by sample size and the computational performance necessary for calculating $n \times (n - 2)/2$ pairwise SNP-SNP interactions for n SNPs. A gene network based on co-expression, protein-protein interaction, and/or functional annotations such as GO/KEGG can shed light on gene-gene interactions".

Discussion part: The sentences like this should be rephrased and re-written: "We calculated the cumulative allele frequency (CAF) of low frequency (allele frequency < 0.05) LoF mutation of 43 IBD-susceptibility genes with eQTL (Supplementary Table 1) in gnomAD database and showed a LoF CAF of 0.1, indicating that an individual has a possibility of 0.1 to carry at least a LoF allele at the 43 IBD-susceptibility genes. Why there can be such a common LoF CAF in a population. The gene expression network in the healthy adults and paediatric can be a explain".

Response: Thanks for your suggestion. These sentences have been re-phrased and re-written [line: 348-354, 300-310].

Discussion part: It would be helpful to give a brief overview of what is already known about *RPL3* to indicate the novelty of the authors findings.

Response: Thanks for your suggestion. We add a discussion of the functional role of *RPL3*. *RPL3* encoded ribosomes that catalyze protein synthesis and is implicated in a number of biological processes. Numerous pieces of evidence indicate that a subset of ribosomal proteins regulate the cell cycle and apoptosis². Moreover, *RPL3* interacts with *DUOX2* that promotes the progression of colorectal cancer cells³. In colorectal cancer cells, knockdown of *DUOX2* inhibits invasion and migration that can be reversed by the overexpression of *RPL3*. The loss of *RPL3* plays an important role in inhibition of cell proliferation upon exposure to Actinomycin D (a widely used anti-cancer drug)⁴. *RPL3* is indicated as a key gene in the regular center of network. Collectively, *RPL3* gene may be a therapeutic target that worth for an attention. [line: 325-347]

Discussion part: Even though the authors reveal genes such as *STAT3*, *ZFP36L1*, and *TET2*, along with *RPL3* as central regulators in health, they do not mention what is the biological meaning of this finding or its relevance.

Response: Thanks for your suggestion. We have added a discussion of these genes. [line:316-325]. We replaced the *ZFP36L1/TET2* by the *PAPK7/RNASET2* genes that play a more important role in the network center. These genes are known to play important role in the regulation of IBD-related function. *STAT3* is a mediator gene that regulate innate and adaptive immunity¹⁰. The loss of *STAT3* in immune cells caused severe inflammation¹⁰. *PARK7* was shown to regulate the IBD-related inflammation in vitro and in vivo¹¹. *RNASET2* expression decreases in response to T cell activation¹². Overexpression of *RNASET2* significantly reduced IFN- γ secretion¹². The *STAT3* and *RNASET2* were suggested to be potential therapeutic targets in the treatment of IBD^{12,13}, as well as *PARK7* for gut-brain axis¹⁴. [line: 318-325]

Reviewer #3 (Comments to the Authors (Required)):

Liu et al 2023 - Life Science Alliance

In this study the authors set out to examine the expression of key IBD susceptibility gene signatures at single cell level using publicly available data. In particular, they generate a catalogue of IBD susceptibility loci candidates from a number of studies and examine expression of this within one of the prominent cell atlases of human (both adult and paediatric) intestine. The authors conclude that the 'majority' of IBD susceptibility loci are found in monocytes in IBD. As detailed below, although the manuscript is easy to follow and the data presented in a clear and concise manner, there are major concerns about the approach used by the authors which very much undermines the conclusions reached.

General and major comments

1. The main concern here, is the imputed public datasets that have been used. The Elmentaite 2021 study by the Teichmann group was one of the first cell atlases of human intestine and has a large number of cells 428,000. However, these are

tremendously variable, which introduces significant confounding variable in the analysis. For example, tissue site (11 distinct locations in the adult samples) and developmental stage (e.g. foetal/paediatric and adult). Most importantly, there are only 7 IBD samples analysed, all of which derive from paediatric samples. How much phenotypic information is known on the paediatric samples? E.g. age of onset, are these VEOIBD, which is a discrete entity, etc. Similarly, what is the ethnicity of the different datasets used in this study, particularly the paediatric patients as we know ethnicity will affect IBD susceptibility traits.

2. For this study to draw the conclusions it currently has, analysis would need to be done across multiple studies to demonstrate the validity of the authors' conclusions that 'majority of IBD susceptibility loci are specifically expressed in monocytes'. For example, the Kong et al. study by the Xavier group in which 46 Crohn's disease samples were compared with 25 healthy controls would be an ideal resource with which to perform this analysis (PMID: 36720220). This is important because paediatric onset IBD represents between 10-25% of all IBD, and the genetic susceptibility underpinning this is likely to be subtly different to adult onset IBD (e.g. PMID: 22543157). As a result, using a small number of paediatric IBD samples matched to adult GWAS data is not the most appropriate paired analysis to be performed.

Specific comments

5. The claim that "The density and cluster of IBD-susceptibility genes network is lower in paediatric IBD patients than in healthy participants" could simply reflect the low power (n=7) of these data.

Response to General and major comments 2, and Specific comments 5: These are important comments. A replication is necessary. We added a table (table 1) that presents a concise overview of the scRNA dataset used for discovery in previous manuscript and an additional scRNA dataset used for replication in revised manuscript [line: 108-118]. Based on the resource you provided, we validated the main results using this additional dataset consisting of 46 samples from individuals with Crohn's disease samples and 25 healthy individuals [methods: 114-118; results: 200-207, 234-236, 242-246, 255-267]. Thank you for these comments.

Specific comments

1. It is unclear how the graph in Figure 1A supports the conclusion that the 'majority of IBD susceptibility loci are specifically expressed in monocytes'. As the authors state themselves, this enrichment analysis equally shows that the IBD gene signature is expressed by Th1 cells and ILC3. Data supporting the author's conclusion is not shown in this figure. For instance, what proportion of the 232 loci were enriched in each of the cell types presented in Fig 1A? How was this weighted, e.g. is the variance explained by a small number of SNPs?

Response: We apologize for the lack of clarity in our previous result of cell specificity expression. Skene et al.'s study^{15,16} designed the EWCE algorithm (Expression Weighted Celltype Enrichment) to indicate the cell type significant enrichment the expression of disease-related genes. This method compares the average expression of genes related to the disease with the background distribution of all genes, randomly sampled. Since the weighted expression was calculated based on the proportion of mean expression among all cell types, it is possible that a small proportion of cell types exhibited a high level of expression for some genes. Put simply, monocytes are not the only one cell type that enriched the expression of IBD genes. Given that monocytes, Th1 cells, and ILC3 cells exhibited a significant proportion of gene expression related to IBD, it can be inferred that other cell types expressed only a minor proportion of these genes. When comparing the gene expression distribution in different cell types to the background, we cannot observe many cell types expressed IBD genes much more than other expressed genes. We added a sentence "A threshold of FDR P-value < 0.05 is used to indicate the cell type that enriched the expression disease-related genes." to the EWCE method [line: 128-130].

3. The data presented in Figure 1B are unclear. What exactly is being shown here? Proportion of ILC3, Th1 cells and monocytes of the entire dataset? Is this meaningful? The authors talk about cell numbers in the text but only shown proportion data. Given monocytes are the focus for analysis, the absolute numbers of these cells in each dataset, in each location and developmental stage, should be presented. Have the monocytes from multiple anatomical locations been pooled for example?

Response: Thanks for your suggestion. We displayed the cell numbers in figure 1B and added a table that outlines the details of case/control, tissue, and their respective cell numbers (Table 1).

4. The rationale for using the current approach is not clear, specifically, why would IBD loci necessarily be expected to act in a network? Would it have been better to look at networks that were different between groups and overlay loci to this?

Response: Nice comment. We are also interested in determining whether the IBD loci act in a network initially. We showed that the majority of IBD susceptibility genes are dosage-sensitive compared with loss of function tolerant genes, known dosage-sensitive genes, and background genes via strictness measure developed in our previous study, as well as the Z-score of the ratio of expected to observed loss of function mutations obtained from the gnomAD database. This result is somehow opposite to the eQTL that significantly modify the level of susceptibility gene expression and also opposite to the common cumulative allele frequency of loss of function mutation at these susceptibility genes. The common SNP with high risk to IBD and with eQTL was believed to affect the disease through expression alteration. How a large proportion of individual with the effective allele exhibit healthy phenotypes? Of course, we can explain this phenomenon through the combined impact of numerous loci with minimal effects, utilizing an additive model, such as polygenic risk score (PRS) model. However, in this view, how can we explain the common cumulative allele frequency of loss of function mutations that result in a reduction of at least 50% in gene dosage? Given the established understanding of gene-gene interaction in biological processes, it is possible that there exists a regulatory network governing a subset of genes that are crucial for IBD. Our study aims to demonstrate the correlation between network state and disease progression.

Minor comments

1. There are a few controversial statements in the introduction that required explanation / citation e.g. "Genetics is the primary driver of IBD with an estimated heritability of 0.75 (CD) and 0.67 (UC) from twin cohort". The authors cite one study to support this statement. The changes in IBD incidence alone in newly-industrialised countries shows that genetics alone cannot be responsible as the primary driver of IBD

4. "These loci explained 13.1% and 8.2% of the disease susceptibility variance for CD and UC, respectively". Citation required

Response to 1 and 4: We have added the citations for the heritability and susceptibility variance in the revised manuscript. The sentence, "Genetics is the primary driver of IBD", has been revised to "Genetic factor plays a critical role in IBD."

2. "The majority of IBD risk loci share consistent effect in CD and UC, and also in European and non-European populations". But these will also be different in many important areas, e.g. *IL23* versus *NOD2*

Response: Thanks for specifying the relationship between *IL23* and UC, as well as the association between *NOD2* and CD. We revised supplementary table 1 and listed the genes associated with IBD, CD and UC separately.

3. "As the GWAS strategy hypothesized that loci are independent, the absent heritability could be due to gene interactions". But could also be due to other factors, e.g. LD, epigenetics etc.

Response: Thank you for offering expert insights into the factors contributing to the phenomenon of missing heritability. We add a supplementary cause of missing heritability, including incomplete linkage disequilibrium (LD) between causal variants and genotyped SNPs¹⁷, missing variant with small effect¹⁸, missing rare variant with large effect¹⁸, structure variant poorly captured by existing arrays¹⁸, and epigenetic modifications¹⁹.

References

1. Lek, M. *et al.* Analysis of protein-coding genetic variation in 60,706 humans. *Nature* **536**, 285–291 (2016).
2. Castro, M. E., Leal, J. F. M., Lleonart, M. E., Ramon Y Cajal, S. & Carnero, A. Loss-of-function genetic screening identifies a cluster of ribosomal proteins regulating p53 function. *Carcinogenesis* **29**, 1343–1350 (2008).
3. Zhang, X. *et al.* DUOX2 promotes the progression of colorectal cancer cells by regulating the AKT pathway and interacting with RPL3. *Carcinogenesis* **42**, 105–117 (2021).

-
4. Russo, A. *et al.* Regulatory role of rpL3 in cell response to nucleolar stress induced by Act D in tumor cells lacking functional p53. *Cell Cycle* **15**, 41–51 (2016).
 5. Salas, A. *et al.* JAK–STAT pathway targeting for the treatment of inflammatory bowel disease. *Nat Rev Gastroenterol Hepatol* **17**, 323–337 (2020).
 6. Yin, J. *et al.* The role of hypoxia-inducible factor 1-alpha in inflammatory bowel disease. *Cell Biol Int* **46**, 46–51 (2022).
 7. Chen, L. *et al.* The role of Th17 cells in inflammatory bowel disease and the research progress. *Front Immunol* **13**, 1055914 (2023).
 8. Chulkina, M., Beswick, E. J. & Pinchuk, I. V. Role of PD-L1 in gut mucosa tolerance and chronic inflammation. *Int J Mol Sci* **21**, 9165 (2020).
 9. Saraav, I. *et al.* Chronic *Toxoplasma gondii* infection enhances susceptibility to colitis. *Proceedings of the National Academy of Sciences* **118**, e2106730118 (2021).
 10. Fu, X. Y. STAT3 in immune responses and inflammatory bowel diseases. *Cell Res* **16**, 214–219 (2006).
 11. Lippai, R. *et al.* Immunomodulatory role of Parkinson’s disease 7 in inflammatory bowel disease. *Sci Rep* **11**, 14582 (2021).
 12. Biener-Ramanujan, E. *et al.* Diagnostic and therapeutic potential of RNASET2 in Crohn’s disease: Disease-risk polymorphism modulates allelic-imbalance in expression and circulating protein levels and recombinant-RNASET2 attenuates pro-inflammatory cytokine secretion. *Front Immunol* **13**, (2022).
 13. Chen, J. *et al.* Therapeutic targets for inflammatory bowel disease: proteome-wide Mendelian randomization and colocalization analyses. *EBioMedicine* **89**, (2023).
 14. Pap, D., Veres-Székely, A., Szebeni, B. & Vannay, Á. PARK7/DJ-1 as a Therapeutic Target in Gut-Brain Axis Diseases. *Int J Mol Sci* **23**, (2022).
 15. Skene, N. G. & Grant, S. G. N. Identification of vulnerable cell types in major brain disorders using single cell transcriptomes and expression weighted cell type enrichment. *Front Neurosci* **10**, 16 (2016).
 16. Skene, N. G. *et al.* Genetic identification of brain cell types underlying schizophrenia. *Nat Genet* **50**, 825–833 (2018).
 17. Yang, J. *et al.* Common SNPs explain a large proportion of the heritability for human height. *Nat Genet* **42**, 565–569 (2010).

-
18. Manolio, T. A. *et al.* Finding the missing heritability of complex diseases. *Nature* **461**, 747–753 (2009).
 19. Trerotola, M., Relli, V., Simeone, P. & Alberti, S. Epigenetic inheritance and the missing heritability. *Hum Genomics* **9**, 17 (2015).

May 21, 2024

RE: Life Science Alliance Manuscript #LSA-2023-02394-TR-A

Dr. Hankui Liu
BGI Group
Yantian road, BGI
Shenzhen 518000
China

Dear Dr. Liu,

Thank you for submitting your revised manuscript entitled "A broken network of IBD-susceptibility genes in the monocytes of IBD patients". We would be happy to publish your paper in Life Science Alliance pending final revisions necessary to meet our formatting guidelines.

- please address Reviewer 1's remaining comments, which are relevant
- please be sure that the authorship listing and order is correct
- please add a Summary Blurb/Alternate Abstract to our system
- please add the Twitter handle of your host institute/organization as well as your own or/and one of the authors in our system
- please note that the titles in the system and on the manuscript file must match
- please consult our manuscript preparation guidelines <https://www.life-science-alliance.org/manuscript-prep> and make sure your manuscript sections are in the correct order
- the Author Contribution selected for Xi Su doesn't qualify them for authorship. Please either update the contributions in our system and the Author Contributions section of the manuscript or let us know if the author should be removed.

A. FINAL FILES:

B. MANUSCRIPT ORGANIZATION AND FORMATTING:

Sincerely,

Reviewer #1 (Comments to the Authors (Required)):

Liu et al have responded adequately to the comments I made in my original review. However, there are several problems that still need attention.:

1. The authors continue to show in Figure 1A that ILC3 cells express high levels of CD-associated genes but indicate in the text that new studies show that this only applies to pediatric patients, not adult patients. Since CD is far more common in adults than pediatric patients the figure is misleading. I suggest it be replaced with a figure derived from data in adults or that data from adults and pediatric patients be included. This comment also takes into consideration the fact that ILC3 cell numbers are actually decreased on pediatric CD patients compared to healthy pediatric patients.
2. The authors use a gaussian graphical model to show that networks of susceptibility genes are disrupted in CD and this leads to poor regulation of gene activity. This central conclusion is based on identification of "connections" between susceptibility genes (or lack thereof). For this reason the authors need to do a better job to explain how these connections are actually established and how they relate to actual experimental data relating to gene interaction. Given that the raw data necessarily consists of information about gene expression the reader must understand how in a particular cell one can convincingly use known information about how the fraction of such expression attributable to a given gene would affect expression or function of another gene previously attributed to the latter. The fact that NOD2 and ATG16L1 do not appear in the CD networks casts doubt on the real world applicability of the data.
3. The above comment applies to the identification of RPL3 as hub gene and target of therapy. The low odds ratio and association with other diseases suggest a minor and disease non-specific relation between this gene and CD. The very basic nature of its function suggests it would be a poor target of therapy. In any case, studies of KI mutations of RPL3 are necessary to determine if they affect gut inflammation in mice.

Reviewer #2 (Comments to the Authors (Required)):

The authors have responded to all the comments with clear explanations and have made the necessary changes to the manuscript

Reviewer #1 (Comments to the Authors (Required)):

Liu et al have responded adequately to the comments I made in my original review. However, there are several problems that still need attention.:

1. The authors continue to show in Figure 1A that ILC3 cells express high levels of CD-associated genes but indicate in the text that new studies show that this only applies to pediatric patients, not adult patients. Since CD is far more common in adults than pediatric patients the figure is misleading. I suggest it be replaced with a figure derived from data in adults or that data from adults and pediatric patients be included. This comment also takes into consideration the fact that ILC3 cell numbers are actually decreased in pediatric CD patients compared to healthy pediatric patients.

Response: Thank you for the suggestion. We added a panel for the result of adult CD in Fig 1A and Fig 1B, respectively. The new Fig. 1A is a better presentation of result than the combination of the previous one and Fig. S1. About Fig. 1B, thank you for indicating the decrease in ILC3 cell numbers in pediatric CD patients compared to healthy pediatric. The decrease in the number of ILC cells was also observed in adult CD patients compared to healthy adults (response Fig.1B). Considering the facts that no statistical evidence for an association of ILC in adult CD [line:103], and Th1 cells was not identified in the study of Kong et al. [line:102], we retained the result of monocytes in Fig.1B only (manuscript Fig.1B).

Fig.1B

2. The authors use a gaussian graphical model to show that networks of susceptibility genes are disrupted in CD and this leads to poor regulation of gene activity. This central conclusion is based on identification of "connections" between susceptibility genes (or lack thereof). For this reason the authors need to do a better job to explain how these connections are actually established and how they relate to actual experimental data relating to gene interaction. Given that the raw data necessarily consists of information about gene expression the reader must understand how in a particular cell one can convincingly use known information about how the fraction of such expression attributable to a given gene would affect expression or function of another gene previously attributed to the latter. The fact that NOD2 and ATG16L1 do not appear in the CD networks casts doubt on the real world applicability of the data.

3. The above comment applies to the identification of RPL3 as hub gene and target of therapy. The low odds ratio and association with other diseases suggest a minor and disease non-specific relation between this gene and CD. The very basic nature of its

function suggests it would be a poor target of therapy. In any case, studies of KI mutations of RPL3 are necessary to determine if they affect gut inflammation in mice.

Response: Nice comments. These two comments point to an important question, what is a gene-gene connection? Here, I combined the responses for the comments 2 and 3. For the first question, how these connections are actually established? We used Pearson's correlation coefficient (r) to estimate the connection of two genes from their expression. As we know there is a network among genes, the correlation coefficients are not independent, which means a connection with a correlation coefficient, such as 0.1, may be a marginal effect caused by some other real, strong connections. So, we cannot use a simple threshold to indicate the real (strong) connection, such as $r > 0.2$. In our study, we used a partial correlation coefficient, ρ , to indicate the true correlation coefficient adjusted by other genes. Gaussian graphical model (GGM) was used to indicate the connection with a $\rho \neq 0$. In some cases, we can simply use the Fisher's transformation, $z = \frac{1}{2} \log\left[\frac{1+\rho}{1-\rho}\right]$; $z \sim N\left(0, \sigma^2 = \frac{1}{n-3}\right)$, and threshold of $P < 0.05$ or FDR to indicate the connection with a $\rho \neq 0$.

For the second question, how they relate to actual experimental data relating to gene interaction? We simply employed a database (STRING, v11.5, <https://cn.string-db.org/>) of protein-protein interaction (PPI) as a truth set to estimate the proportion of truth gene-gene connections for our study. The PPI database identified a number of 11,938,498 interactions from a space of sample of $\frac{n(n-1)}{2}$ interactions, where n refers to the number of 19,566 protein coding genes. We defined there is 11,938,498 truth interactions from 191,404,395 possible interactions, the proportion of truth interactions is 0.062, that means when we randomly choose two genes from all genes for 100 times, there is approximately 6 truth gene-gene interactions. The GGM identified 138 gene-gene connections from 232 genes in scRNA of healthy adults. Out of these, 41 connections were identified in PPI database. The proportion of "real" connections is 30%. A binomial test (P -value= 1.57×10^{-17}) indicates the performance of GCM is significantly higher when compared to a null hypothesis that randomly choose connections from the 232 genes. Considering the

gene-gene connections were identified from scRNA of gut via correlation coefficients, and the protein-protein interactions were identified from bulk data via a combined score of experimental data and array, we treat this analysis as an estimation but not a validation. We have added an explanation for the how these connections are actually established into the manuscript [lines: 139-142, 340-345,366-372].

For the absence of *NOD2* and *ATG16L1* from CD networks, there are two explanations. First, *NOD2* and *ATG16L1* are not especially expressed in monocytes. We showed an association of monocytes and the majority of IBD-related genes, but not for all genes, including *NOD2* and *ATG16L1*. For genes with a low proportion of expressed cells, it has been excluded from network building. Second, *NOD2* has a monogenic role in early onset IBD (doi: 10.1093/ecco-jcc/jjac124; <https://doi.org/10.1038/s41598-021-84938-8>). Our previous study (<https://doi.org/10.3389/fnins.2023.1116087>) indicated the expression strictness of pathogenic genes are differed from the susceptibility genes in ALS, indicating the pathogenic genes are tolerant to expression alternation. The pathogenic genes with causal effect may not work with the susceptibility genes that play through risk, where a proportion of 18% susceptibility genes were modified by eQTL. In contrast, the hub gene, *RPL3*, presents a low odds ratio of 1.12 in IBD risk. In CD patients, genes connected to *RPL3* or through *RPL3* are gone. From this point of view, the *RPL3* gene is important to maintain the normal network. After carefully reviewing the mean of therapy target, we instead it by the regulatory target to avoid a misunderstanding for reader [line:247]. Finally, by considering the limitations of our study, it is a challenge to include a KI/KO experiment for the role of *RPL3* in our present manuscript. It will be interesting to investigate the role of *RPL3* by regulating its dosage in our further study.

Reviewer #2 (Comments to the Authors (Required)):

The authors have responded to all the comments with clear explanations and have made the necessary changes to the manuscript

Response: Thank you very much for your positive feedback.

June 17, 2024

RE: Life Science Alliance Manuscript #LSA-2023-02394-TRR

Dr. Hankui Liu
BGI Group
Yantian road, BGI
Shenzhen 518000
China

Dear Dr. Liu,

Thank you for submitting your Resource entitled "A broken network of susceptibility genes in the monocytes of Crohn's disease patients". It is a pleasure to let you know that your manuscript is now accepted for publication in Life Science Alliance. Congratulations on this interesting work.

DISTRIBUTION OF MATERIALS:

Again, congratulations on a very nice paper. I hope you found the review process to be constructive and are pleased with how the manuscript was handled editorially. We look forward to future exciting submissions from your lab.

Sincerely,
